# Risk Factors for Pulmonary Embolism in Individuals Infected with SARS-CoV2—A Single-Centre Retrospective Study

**DOI:** 10.3390/biomedicines12040774

**Published:** 2024-04-01

**Authors:** Alexandra Herlo, Adelina Raluca Marinescu, Talida Georgiana Cut, Ruxandra Laza, Cristian Iulian Oancea, Diana Manolescu, Elena Hogea, Tamara Mirela Porosnicu, Suzana Vasilica Sincaru, Raluca Dumache, Sorina Ispas, Andreea Nelson Twakor, Maria Nicolae, Voichita Elena Lazureanu

**Affiliations:** 1Department XIII, Discipline of Infectious Diseases, Victor Babes University of Medicine and Pharmacy Timisoara, 2 Eftimie Murgu Square, 300041 Timisoara, Romania; alexandra.mocanu@umft.ro (A.H.); adelina.marinescu@umft.ro (A.R.M.); talida.cut@umft.ro (T.G.C.); laza.ruxandra@umft.ro (R.L.); lazureanu.voichita@umft.ro (V.E.L.); 2Department XIII, Discipline of Pneumology, Victor Babes University of Medicine and Pharmacy Timisoara, E. Murgu Square, Nr. 2, 300041 Timisoara, Romania; oancea@umft.ro; 3Center for Research and Innovation in Precision Medicine of Respiratory Diseases (CRIPMRD), Victor Babes University of Medicine and Pharmacy Timisoara, E. Murgu Square, Nr. 2, 300041 Timisoara, Romania; 4Department XV, Discipline of Radiology, Victor Babes University of Medicine and Pharmacy Timisoara, E. Murgu Square, Nr. 2, 300041 Timisoara, Romania; dmanolescu@umft.ro; 5Department XIV, Discipline of Microbiology, Victor Babes University of Medicine and Pharmacy Timisoara, E. Murgu Square, Nr. 2, 300041 Timisoara, Romania; hogea.elena@umft.ro; 6Doctoral School, Victor Babes University of Medicine and Pharmacy Timisoara, 300041 Timisoara, Romania; mirela.porosnicu@umft.ro; 7Intensive Care Unit, Victor Babes Clinical Hospital for Infectious Diseases and Pneumology, 300041 Timisoara, Romania; 8Emergency Institute for Cardiovascular Diseases and Transplant, Strada Gheorghe Maricescu, 540327 Targu Mures, Romania; 9Department of Forensic Medicine, Bioethics, Medical Ethics and Medical Law, “Victor Babes” University of Medicine and Pharmacy, 300041 Timisoara, Romania; raluca.dumache@umft.ro; 10Department of Anatomy, Faculty of General Medicine, “Ovidius” University, 900470 Constanta, Romania; sorina.ispas@365.univ-ovidius.ro; 11Department of Internal Medicine, County Clinical Emergency Hospital of Constanta, 900591 Constanta, Romania; andreeanelsont@gmail.com; 12Department of Pediatrics, County Clinical Emergency Hospital of Constanta, 900591 Constanta, Romania; dr.nicolae_maria@yahoo.com

**Keywords:** COVID-19, SARS-CoV-2, thrombosis, mortality, pulmonary embolism

## Abstract

The emergence of SARS-CoV2 has presented itself as a significant global health crisis. The prevalence of thrombotic events is known to be high in these patients, affecting various organ systems, sometimes leading to cutaneous thrombosis, pulmonary embolism (PE), stroke, or coronary thrombosis. The available evidence suggests that thromboembolism, hypercoagulability, and the excessive production of proinflammatory cytokines play a significant role in the development of multiorgan failure. **Methodology:** This retrospective single-centre study was conducted at “Victor Babes” University of Medicine and Pharmacy from Timisoara, Romania, involving a total of 420 patients diagnosed with COVID-19. We separated them into a CONTROL group that included 319 patients, and an intervention group (PE) with 101 patients that, subsequent to infection with the virus, developed pulmonary embolism. The study included the reporting of demographic data, laboratory findings, and comorbidities. **Results:** Out of a total of 420 patients, 24% experienced pulmonary embolism, while 21.42% died. Arterial thrombotic events were found to be associated with factors such as age, cardiovascular disease, levels of white blood cells, D-dimers, and albumin in the blood. The findings of the study indicate that there is an independent association between pulmonary thrombosis and hypertension (odds ratio (OR): 1.1; 95% confidence interval (CI): 0.7 to 1.7; *p* = 0.6463), cancer (OR: 1.1; 95% CI: 0.6 to 2.3; *p* = 0.6014), and COPD (OR: 1.2; 95% CI: 0.6 to 2.3; *p* = 0.4927). On the other hand, there is a stronger correlation between PE and obesity (OR: 2.8; 95% CI: 1.7 to 4.6; *p* < 0.0001), diabetes (OR: 3.3; 95% CI: 2 to 5.3; *p* < 0.0001), and dyslipidemia (OR: 3.6; 95% CI: 2.3 to 5.8; *p* < 0.0001) in a multivariable regression logistic model. **Conclusions:** Patients diagnosed with severe forms of COVID-19 display a comparable incidence of arterial thrombotic events, which have been linked to poor survival rates.

## 1. Introduction

The emergence of the new severe acute respiratory syndrome—coronavirus 2 (SARS-CoV2)—which is responsible for the onset of coronavirus disease 2019 (COVID-19), has resulted in an unparalleled worldwide health emergency. As of the present time, the number of fatalities attributed to COVID-19 exceeds 6.9 million cases worldwide [1]. 

A significant majority of individuals who test positive for SARS-CoV2 have either moderate or no symptoms at all [2]. However, severe COVID-19 infections show systemic inflammation, endothelial dysfunction, coagulation activation, acute respiratory distress syndrome (ARDS), and multiorgan failure [3]. Evidence of myocardial damage is also seen in at least 25% of severe cases [4,5].

Ongoing research suggests that the higher death rates found in individuals with COVID-19 might be attributed, at least in part, to undetected occurrences of arterial thromboembolic complications such as pulmonary embolism (PE) and pulmonary in situ thrombosis [6,7,8]. The current assessment of these risks is still being investigated, and it depends on the preventive strategies that are in place [9]. PE can lead to life-threatening complications such as acute respiratory failure, cardiogenic shock, and sudden cardiac arrest. These complications can occur rapidly and necessitate immediate medical intervention [10]. While respiratory failure continues to be the primary cause of mortality in cases with moderate or severe COVID-19 infections, there have been several reports supporting cardiovascular consequences and incidents of thromboembolic events [11,12,13].

This study looks at thromboembolism as it is essential for advancing prevention, diagnosis, and treatment strategies, reducing complications, improving quality of life, and addressing the significant public health burden associated with these disorders. We look at the hypercoagulation state that is observed in patients with COVID-19, as this may be caused by a combination of factors including an excessive release of inflammatory cytokines and prolonged stasis [14]. The literature also shows that inflammation may significantly contribute to endothelial damage, platelet activation, and the production of tissue factor-bearing microparticles [15]. 

Our research is also looking at individuals diagnosed with hypercoagulability, as they face a significantly higher chance of experiencing thrombotic events, such as venous thrombosis and arterial thrombosis [16]. 

Although there is a possibility of a fundamental prothrombotic condition, there is a lack of available evidence on the risk of acute arterial thrombotic incidents. The objective of this research is to provide the characteristics and outcomes of 420 COVID-19 hospitalised patients at “Victor Babes” Hospital for Infectious Disease and Pneumophtisiology from Timisoara, Romania, and their risk of developing pulmonary embolism. 

Moreover, as many countries in Europe are facing the 7th wave of the COVID-19 pandemic since September 2023 [17], moving forward from an unprecedented massive admission of patients with consequent pneumonia requiring invasive mechanical ventilation, sometimes leading to ICUs’ saturation (1st wave) of a milder strain of the virus (7th wave), this study also compares the characteristics and the outcomes of the PE group throughout the pandemic so far.

## 2. Materials and Methods

### 2.1. Study Design and Ethics

In this research, we included data from adult symptomatic patients with confirmed COVID-19 who were admitted to “Victor Babes” Hospital for Infectious Disease and Pneumophtisiology from Timisoara, Romania. The study was approved by the institutional ethical committee, and patients provided standard written consent to the use of their data.

The study period covered 1 February 2020 to 30 September 2023. The identification of COVID-19 was achieved using the recommendations provided by the World Health Organization.

This research used a retrospective methodology setting, containing a sample of 420 eligible patients aged 51 years or older from each of the seven pandemic wave groups. These patients were matched based on their symptoms, comorbidities, gender, and the development of thrombotic events. In addition, the period of study includes the months before the COVID-19 vaccination period, as well as after.

The Infectious Disease Clinic affiliated with the “Victor Babes” University of Medicine and Pharmacy, as an auxiliary of the “Victor Babes” Hospital for Infectious Disease and Pneumophtisiology from Timisoara, operates under the laws of the local commission of ethics that approves scientific research that functions in accordance with the International Conference on Harmonization from Helsinki regarding technical requirements for the registration of pharmaceuticals for human use. This research complied with the ethics criteria from the university where the study was developed and was approved by the ethics committee of both institutions on 3 October 2023, with approval number 9148.

### 2.2. Inclusion and Exclusion Criteria

A comprehensive search was performed on both electronic and physical patient records to determine the prevalence of TEP hospital admissions among individuals diagnosed with SARS-CoV-2 infection. 

Multiple journal papers were used to elucidate the process of sample size selection and the criteria for determining an appropriate sample size [18,19,20,21].

We used the criterion sampling method described by Moser et al. [22], which refers to the selection of participants who meet pre-determined criteria of importance. 

We only selected patients that matched the following criteria: ○Aged 51 or older;○Positive diagnosis of a moderate or severe form of COVID-19, confirmed through a RT-PCR test;○Hospitalisation due to acute infection with SARS-CoV2, with at least one of the following comorbidities: hypertension, diabetes, COPD, CAD, heart failure, atrial fibrillation, dyslipidaemia, BMI ≥ 30, and/or cancer history; ○Development of pulmonary embolism;○Vaccination or lack thereof (disregarding what type of vaccine was administered)

We excluded patients that had the following characterises:○An age of 50 or below○Symptomatic and mild forms of COVID-19 infection;○Development of venous thromboembolism. 

The patients were chosen after a thorough evaluation of the inclusion criteria to guarantee the pertinence, practicability, and ethical execution of the study. The needed sample size was estimated by taking into account statistical factors such as the projected effect size, degree of significance, research power, and expected dropout rates [23]. 

In our decision on sampling, we also used phenomenology [24], in which (similar to the concept above) our participants had to meet a predefined criteria. The determining factor was the participant’s criteria with the phenomena being investigated. We wanted our study patients to be investigated based on who had an event (of infection with SARS-CoV2), but who were different in terms of their features and individual experiences. Thus, our phenomenological investigation included individuals that tested positive for COVID-19 but with diverse characteristics such as age, comorbidities, the existence of pulmonary thromboembolism, and the development of a severe form of the infection.

In total, 6879 patients were hospitalised between February 2020 and September 2023 at “Victor Babes” Hospital for Infectious Disease and Pneumophtisiology from Timisoara, Romania. Among them, 529 were admitted to the ICU department, while 1394 died. After applying the inclusion and exclusion criteria, we selected 1812 that were over the age of 51 at the time of hospitalisation. In total, 915 of them had mild forms of COVID-19 infection, and thus were disregarded from the study, leaving us with 897 patients to consider. Out of them, 142 were transferred to other medical centres. From the remaining 755 subjects, 103 developed thrombotic events in locations other than the pulmonary area, and 232 had comorbidities other than the ones listed in our inclusion criteria. The remaining 420 patients were included in the study.

We based our statistical approach on this information and analysed the clinical presentations and various factors that led to the development of PE. Out of the remaining 420 patients with a severe form of COVID-19, in total, 101 individuals progressed to pulmonary embolism after meeting the predetermined inclusion and exclusion criteria. Due to the absence of PE, 319 individuals were assigned to the CONTROL group. 

Figure 1 includes a flowchart describing the steps of this recruitment process

Following this selection criteria, we categorised COVID-19 infection as moderate and severe using the national guidelines. 

According to the Romanian Ministry of Health with regard to the clinical spectrum of SARS-CoV-2 infection, the illness can be classified as follows [25]:○Moderate infection:
SpO_2_ ≥ 92%;Pulmonary lesions on <50% of the surface (as seen imagistically);CRP > 50 mg/dL;Twofold or threefold increased ferritin;Threefold increased D-dimers;Thrombocytopenia with values between 100,000 and 150,000/mcL;Lymphopenia with the range 1000–1500/mcL;Lactate with values double the normal.○Severe infection:SpO_2_ < 90%;PaO_2_/FiO_2_ ratio ≤ 300 mmHg;Respiratory rate over 30 breaths per minute;Lung infiltrates on more than 50% of the surface;CRP >100 mg/dLIncreased ferritin, IL-6, and D-dimers (3 to 4 times higher);thrombocytopenia (less than 100,000/mcL).

WHO mentions that in a severe infection, SpO_2_ is <90% with signs of pneumonia and severe respiratory distress (accessory muscle use; inability to complete full sentences) [26].

We further classified patients according to the pandemic waves, aiming to maintain an approximately similar percentage of patients for each of them. We considered that a number of 60 patients from each period would give significant statistical importance and would offer relevant conclusions that could be applied to the general population. We considered the periods of pandemic wave according to how Romanian Ministry of Health established them [27].

Thus, we included samples of patients for the waves described below [27]:○First wave: January–June 2020 (Wuhan-Hu-1—NCBI Reference Sequence: NC 045512.2 variant was the predominant strain);○Second wave: July–December 2020 (Clade variants S: D614G being the dominant viral strains);○Third wave: January–March 2021 (Alpha B.1.1.7 variant);○Fourth wave: September–November 2021 (Delta B1217.2 variant);○Fifth wave: January–March 2022 (Omicron variant);○Sixth wave: April–November 2022 (BA.2 subvariant of Omicron);○Seventh wave: December 2022–March 2023 (BA.5 subvariant of Omicron).

Based on all the above, it was determined using a convenience sampling method that a minimum of 319 adult patients 51 years or older hospitalised for SARS-CoV-2 infection was sufficient to provide the statistical power needed for the CONTROL group, and the total of 101 patients who developed arterial thromboembolic events was sufficient for the pulmonary embolism (PE) group. 

### 2.3. Study Variables

The variables that were included for analysis were as follows: ○Baseline characteristics of the participants, including age, BMI, gender, demographics, smoking status, comorbidities (diabetes, COPD, CAD, heart failure, atrial fibrillation, dyslipidaemia, hypertension, cancer history), COVID-19 vaccination status, and COVID-19 vaccine type; ○Paraclinical findings such as WBC count, platelets, lymphocytes, GOT, GPT, ALT, ferritin, ERS, CRP, fibrinogen, procalcitonin, D-dimers, and others; ○Clinical findings and disease outcomes such as signs and symptoms at admission, COVID-19 outcomes, disease severity, duration of hospitalisation, ICU admission and duration, Wells and Pesi scores, O_2_ supplementation, and mortality.

### 2.4. Statistical Analysis

The frequency of occurrences was accompanied by a 95% confidence interval (95% CI) and calculated for closed cases, which were defined as patients who were discharged, deceased, or diagnosed with a pulmonary thromboembolic event.

We conducted a univariate logistic regression analysis to examine the impact of these complications on the chance of patient mortality during hospitalisation. The results are shown graphically to illustrate the relationship between risk factors of PE events and different variables. Data analysis was conducted using Statistical Package for the Social Sciences (SPSS v. 29.0).

## 3. Results

### 3.1. Thrombotic Events in COVID-19

In total, 420 individuals were admitted to the hospital, with a median duration of hospitalisation of 13.4 ± 3.6 days (interquartile range: 12–22). Among these patients, 374 (89.05%) were admitted to infectious disease wards, while 46 (10.95%) were admitted to the intensive care unit. Table 1 presents the initial clinical and laboratory characteristics, as well as the treatments administered upon admission to the hospital. Out of the total number of patients, 101 individuals (24%) experienced pulmonary thrombotic events during their hospital stay. 

In comparison with patients who did not develop PE during their hospital stay, patients who did experience such an event were found to be of a higher mean age (71.86 ± 13.27 vs. 67.47 ± 15.52, *p* = 0.329). Additionally, they displayed a greater likelihood of having a history of coronary artery disease (18.81% vs. 15.98%, *p* = 0.144) and being admitted to the intensive care unit (19.8% vs. 7.8%, *p* = 0.006). Additionally, the PaO_2_/FiO_2_ ratio upon admission to the hospital was observed to be lower in patients who experienced thrombotic events, with a median of 254 [155–309] vs. 328 [284–379] (*p* < 0.001). In terms of laboratory variables, it was observed that patients who experienced PE showed higher baseline levels of white blood cells, serum D-dimers, and ferritine, while their albumin levels were comparatively lower (Table 1).

Pulmonary thrombotic events were found to be independently associated with hypertension (odds ratio (OR): 1.1; 95% confidence interval (CI): 0.7 to 1.7; *p* = 0.646), cancer (OR: 1.1; 95% CI: 0.6 to 2.3; *p* = 0.601), and COPD (OR: 1.2; 95% CI: 0.6 to 2.3; *p* = 0.492). On the other hand, there is a stronger corelation between obesity (OR: 2.8; 95% CI: 1.7 to 4.6; *p* < 0.001), diabetes (OR: 3.3; 95% CI: 2 to 5.3; *p* < 0.001), and dyslipidaemia (OR: 3.6; 95% CI: 2.3 to 5.8; *p* < 0.001) in a multivariable regression logistic model (Table 2).

Gender was excluded from our original recruitment approach due to compelling data in the literature indicating that pulmonary embolism is equally prevalent in both men and women [28,29,30,31]. This is supported by the data from our remaining study participants, where 55.45% of men had PE, compared with 44.55% of women with the same condition. The 0.8% disparity between the two genders is insufficient, in our view, to support the idea that one gender is more susceptible to developing PE than the other.

A value greater than 1 implies a higher frequency of an event [32]. Alternatively, <1 signifies a reduced frequency of an event (exposure that provides protection) [33]. Thus, when we interpret these values, we consider the result after the first numerical value (e.g., for the value 1.19 for the cancer variable, we consider a 19% higher chance of developing PE while having cancer when compared with that in the CONTROL group).

Taking all this into account, we can conclude the following: the PE group has 11% higher chances of having hypertension in comparison with the CONTROL group, 19% higher changes for cancer, and 25% higher chances of having COPD as a comorbidity.

The odds ratios for obesity, diabetes and dyslipidaemia can be interpreted as follows: the PE group has 186% greater odds of being obese when compared with the CONTROL group, 233% greater odds of having diabetes, and 267% greater odds of having dyslipidemia. 

Moreover, if *p* < 0.001, the evidence is considered highly statistically significant. This is the case for the odds ratio for obesity, diabetes and dyslipidaemia. In general, a lower *p*-value is preferable, as it suggests that the observed results are less likely to have occurred by random chance.

*p*-Values such as 0.6463, 0.6014, and 0.4927 (hypertension, cancer, and COPD) suggest that there is a probability that the observed results occurred by random chance. 

While the proportion of vaccinated individuals was higher among older patients compared with younger persons in our sample, it is noteworthy that the severity of SARS-CoV-2 infection was much more pronounced among the elderly population (Table 1).

### 3.2. Thrombus Location

We further analysed the patients from the PE group by looking at the thrombus location. Thus, we organised the comorbidities, selected markers, and outcome scores according to the location of the thrombus (central, lobar, segmental, and peripheral). 

The data summarised in Table 3 show that the average age of patients with segmental PE is higher at about 4 years when compared with that of patients in the central PE subgroup (70.44 ± 9.44 vs. 66.5 ± 8.03). The male to female ratio is almost similar for all the subsections, with the exception of lobar where 60% of the patients who developed PE in this area were men. 

The segmental pulmonary embolism accounted for 50.49% of the cases (51 patients), whereas central, lobar, and peripheral PE accounted for 5% (6), 19.8% (20), and 24.75% (25), respectively. 

As it can be seen in Table 2, patients with central artery involvement were of a younger age. Figure 2 also shows that the same patients had fewer underlying medical conditions such as hypertension and dyslipidaemia (six cases each), COPD (two cases), and obesity and cancer (two cases combined). The same patients had lower platelet and D-dimer counts when compared with the other subgroups, but slightly higher Pesi and Wells scores. However, this upper value is statistically insignificant as the difference between the four groups is less than 0.4. 

The peripheral PE group consisted of 25 patients. The most prevalent comorbidities in these patients were hypertension, diabetes and dyslipidaemia (16 cases each). The same co-existing conditions were seen in high numbers in segmental and lobar groups as well (18 and 24 patients with hypertension, 16 and 31 patients with diabetes, and 14 and 29 patients with dyslipidaemia).

Patients in the lobar subcategory registered the highest D-dimer average when compared with the rest of the subgroups (5888 ± 3411.92), while those in the segmental one had the highest average platelets count (243.1 ± 122.4).

Figure 3 shows a forest plot analysis of the thrombus locations analysed in this section based on Wells and Pesi scores. The Wells score was calculated at the beginning of hospitalisation for all the patients. In order to create a forest plot, we carefully chose the Wells score for the 101 patients who were part of the PE group. We next examined the correlation between these scores and the Pesi score, which was determined after these patients had developed PE. We included all the means and standard deviations of these scores and generated a visual summary of the individual thrombus locations and their overall combined effect.

In the context of this study, the weights shown by the green circles refer to the relative magnitude assigned to each thrombus site. These weights are derived based on the Wells and Pesi scores, as well as the sample size. The weight signifies the relative importance of each thrombus location in determining the overall estimate of the effect size.

Thus, the weight of 50.08 assigned to a segmental subgroup indicates that this study has a significant influence on the entire analysis due to its relatively large sample size and the Wells and Pesi scores taken into account during the weighting procedure. By contrast, the lowest weight is given to the central subgroup, having only a value of 6.79.

The effect size (red diamond) of −1.42483 indicates the estimated association of Wells and Pesi scores for each thrombus location used for this analysis. The negative sign suggests a negative impact is associated with the parameters studied. Thus, even though some patients had a low Wells score at the beginning of hospitalisation, they developed PE later on, the overall result of the forest plot showing this negative relationship being −1.42483. 

### 3.3. Mortality within the PE Group

During the hospital follow-up period, the death rate for the PE group was 31.68% (32 patients), as shown in Table 1. These patients had distinct demographic and clinical characteristics that distinguished them from both the individuals who experienced an improved medical condition and those belonging to the CONTROL group. More precisely, the individuals in question were often older, and had a greater prevalence of COPD, diabetes, heart failure, and CAD (Table 1). Additionally, the group had a greater number of ICU admissions and intra-hospital thrombotic events in comparison with those who had complete recovery. The study revealed a significant correlation between mortality and the initial levels of D-dimers, C-reactive protein, and albumin, as shown by the laboratory variables presented in Table 1. 

Moreover, Table 4 shows the relationship between the Wells score, Pesi score, other relevant variables and the outcome of patients from the PE group. 

Table 4 shows that the age group above 71 had the greatest death rate, together with the Wells and Pesi scores. In contrast, individuals within the age range of 51–60 had the most pronounced pulmonary infiltrations, as shown on the CT scan, thus requiring invasive ventilation compared with other age groups (28%). Their death rate closely matched that of those aged 71 and above, which may be partially explained by the extensive lung damage.

Figure 4 and Figure 5 graphically show the relationships between pulmonary imaging, mechanical ventilation, and mortality, as well as, Pessi score and mortality, respectively.

### 3.4. Statistical Analysis of Other Selected Variables

The relationship map shows the association count between selected variables. In this case, we compared the outcome (fatal or discharge) for those with symptoms, the existence of comorbidities, ICU prevalence, and the use of anticoagulants. 

Figure 6 below shows this relationship map. It should be noted that no. 0 stands for NO (comorbidities, use of anticoagulants, symptoms, or ICU needed) and no. 1 stands for YES (comorbidities, use of anticoagulants, symptoms, or ICU needed). The thicker lines between the variables reflect a stronger relationship between the two. 

The Kaplan–Meier analysis revealed that patients who experienced pulmonary thrombotic events displayed a greater likelihood of mortality in comparison with patients from the CONTROL group (25.74% vs. 18.18%; *p* < 0.001) (Figure 7). 

The present analysis strengthens the conclusions of previous investigations about the associations between thrombosis and reduced survival [34,35,36]. This research found that intra-hospital occurrences of venous and arterial thrombotic events, as well as age and albumin levels, were substantially linked with death using a multivariable analysis. The study’s novelty lies in the observation, as indicated by Kaplan–Meyer survival analysis, of the divergence of the curves within the initial 10 days following hospitalisation. This finding suggests that thrombosis may serve as an early indicator of unfavourable prognosis.

In Table 5, we present a series of correlations between the outcome and symptoms, ICU, comorbidities and anticoagulant therapy. 

A correlation coefficient of 0.673 indicates a moderate positive linear relationship between the outcome and the admission to the ICU. 

In statistics, correlation coefficients range from −1 to 1, where [37] the following apply:○1 indicates a perfect positive linear relationship;○−1 indicates a perfect negative linear relationship;○0 indicates no linear relationship.

A correlation coefficient of 0.673 (outcome-ICU) falls between 0 and 1, suggesting a positive association. The closer the correlation coefficient is to 1, the stronger the positive linear relationship [38].

A correlation coefficient of 0.563 between outcome and symptoms suggests a positive linear relationship as well. Both variables tend to go up in response to one another, and thus the relationship is strong.

The correlation value of 0.161 suggests a positive relationship between the outcome variable and comorbidities. Nevertheless, it is considered very weak, indicating that the correlation between the two variables is not significant.

The correlation coefficient between the outcome and anticoagulant medication is −0.113, suggesting a modest negative linear association between the two variables. This observation suggests that there is an inverse relationship between the two variables. 

The matrix of chosen variables is shown in Table 6. It serves as a visual representation, whereby each column is assigned a correlation coefficient. The numerical value of 1 indicates a strong correlation between variables, whereas a value of 0 signifies a neutral link, and a value of −1 signifies a weak relationship [38].

It is worth mentioning that there are mild associations between comorbidities and anticoagulant treatment (−0.038), whereas large associations are seen between comorbidities and symptoms (0.743), as well as between anticoagulant therapy and symptoms (0.165). The most robust correlation is shown to be between ICU admission and the emergence of symptoms, with a coefficient of 0.814.

Figure 8, Figure 9, Figure 10 and Figure 11 show random-effects meta-regression plots that were used to explore the relationship between continuous predictor variables (comorbidities, anticoagulant therapy, ICU admission, and symptoms) and the effect sizes (outcomes) observed in the patients from the CONTROL and PE groups.

The middle line represents the predicted effect, the left red circle refers to the CONTROL group’s outcome, the right red circle refers to the PE group’s outcome, and the smaller blue circles refer to the variable discussed for each corresponding group.

As it can be observed, in three out of four instances (Figure 8, Figure 10 and Figure 11), there is a positive linear relationship between the two variables analysed, but in one case (Figure 9), there is a negative relationship. 

This study demonstrates that individuals admitted to the hospital for COVID-19 display incidents of arterial ischemic events, which serve as indicators of unfavourable prognosis. 

### 3.5. Comparison of COVID-19 Pandemic Waves for PE Group

Table 7 presents the dynamic comparison of patients included in the PE group who had COVID-19 infection throughout the pandemic.

During the fourth wave in Romania, when the Delta B1217.2 variant of the SARS-CoV-2 virus was predominant, there was a notable increase in the number of patients with a severe form of the infection, as well as mortality amongst these individuals [39]. This increase was statistically significant, as can be seen in Figure 12.

The above results confirm the official statistics provided by the Romanian National Institute of Public Health, when Romania entered its fourth wave of the pandemic in early September 2021, reaching its highest point in the final week of October [40]. Table 7 shows that the Delta B1217.2 variant has put a significant strain on Romania’s healthcare system, leading to the emergence of a fourth wave of the pandemic. It confronted obstacles in effectively controlling transmission and mitigating the consequences of the disease, encompassing issues related to healthcare infrastructure as well as demographic and societal factors [41]. The literature also shows that severe clinical forms had a markedly higher median duration of disease prior to hospitalisation compared with the mild and moderate forms [42].

## 4. Discussions

This retrospective single-centre study provides a comparison between features of COVID-19 patients and those who also had pulmonary embolism, as well as an overview of the inflammatory and biochemical markers that occurred during their hospital stay. Our investigation supports the literature and revealed that age, obesity, diabetes, and hypertension did present a significant correlation with an increased occurrence of pulmonary embolism [43]. 

It has been shown that the most frequently seen symptoms across all seven waves were fever, cough, and fatigue [44]. Less frequently seen concomitant symptoms were nasal discharge, cephalalgia, and gastrointestinal manifestations [45]. In our study, the increased mortality risk among elderly patients hospitalised for COVID-19 was found to be particularly pronounced during the first, third, and fourth waves of the pandemic. 

Improving understanding of the thromboembolic risk associated with COVID-19 will contribute to the modification of diagnostic approaches and inform the planning and implementation of randomised controlled studies pertaining to the prevention of venous thromboembolism [46].

Thrombotic complications have been proposed as a significant contributing factor to decreased survival rates [47]. Based on clinical investigations, it has been observed that there is a significant occurrence of arterial thromboembolism among individuals infected with COVID-19 [48]. Furthermore, a limited number of postmortem studies have indicated a correlation between lung circulation thrombosis and unfavourable patient outcomes [33]. In contrast, researchers have examined the occurrence of arterial thrombosis in limited sample sizes, and it seems to be significantly less prevalent compared with venous thromboembolism [49]. A recent review indicated that the prevalence of arterial thrombosis ranged from 2.8% to 3.8% [50]. Although there exist reliable data regarding the correlation between hypercoagulability and mortality, the influence of venous and arterial thrombosis on survival remains uncertain [51]. 

Pulmonary arterial hypertension is a chronic condition characterised by a gradual alteration in pulmonary arteries, including pathological remodelling, vasoconstriction, and thrombosis. There is a consistent observation of changes in haemostasis, coagulation, and platelet activation among individuals with pulmonary arterial hypertension (PAH) [52]. Microparticles originating from platelets, inflammatory cells, and the endothelium have gained significant recognition as a prominent indicator in several cardiovascular disorders, such as pulmonary arterial hypertension [53]. Pulmonary arterial hypertension involves heightened vascular resistance and the gradual decline in the pulmonary circulation [54].

The occurrence of thrombocytopenia has been seen in some patients who have been diagnosed with pulmonary arterial hypertension (PAH); however, the precise aetiology of this phenomenon remains unknown [55]. The causes of thrombocytopenia are still not fully understood, since it is unsure whether it is caused by anomalies in the production of platelets, the removal of platelets in the bloodstream, or a combination of both [56]. Kobayashi et al. suggested a mechanism that involves pulmonary microangiopathy, in which platelets experience shearing forces as they travel through fibrin clots and plexiform lesions [57]. 

Alterations in haemostasis and coagulation have been frequently found in patients with COVID-19 [43]. While warfarin is often prescribed, there is a scarcity of randomised studies that have thoroughly investigated its possible therapeutic benefits [58]. The recent use of innovative oral direct factor Xa and thrombin inhibitors represents a compelling approach to evaluate the advantages and disadvantages of anticoagulant treatments in clinical studies including patients with PAH [59]. There have been several instances of excessive platelet activation documented, and there are valid theoretical justifications for postulating that this excessive activation of platelets plays a role in the evolution of diseases [60].

The available evidence about the efficacy of anticoagulant medicine as a therapy for pulmonary thrombolism is limited, as shown by the guidelines [61]. The present evidence mostly comprises epidemiological data that provide conflicting results, with the exception of the findings from a supplemental study of a pre-existing treatment trial undertaken before the introduction of drugs targeting pulmonary arterial hypertension [62]. Cohort research was conducted on patients with connective tissue disease-associated pulmonary arterial hypertension (CTD-PAH), mostly scleroderma, which indicated that those who were undergoing anticoagulation medication with vitamin K antagonists (VKAs) had superior survival outcomes compared with those who did not receive anticoagulant therapy [63].

Given that pulmonary embolism is a prevalent and avoidable cause of mortality in hospital settings, the implementation of thromboprophylaxis becomes an essential element in the management of patients who are admitted [64]. This holds true even in the context of COVID-19. 

Thromboembolisms that are triggered by a provoking cause have been shown to have associations with established risk factors, many of which are of a temporary nature [65]. On the other hand, unprovoked thromboembolisms may suggest an elevated propensity for clot formation. The majority of deep vein thromboses seen in the emergency room are classified as unprovoked and have a higher likelihood of recurring compared with provoked DVTs [66]. Specifically, unprovoked thromboembolism has a recurrence rate of 15% during the subsequent 12 months, while provoked PE has a recurrence rate of 5% [67]. It is noteworthy that a significant proportion, around 80%, of individuals who develop PE possess at least one recognised risk factor, and in many cases, numerous risk factors are present [68].

Frequently, individuals with hereditary thrombophilias remain uninformed about their illness until they receive a diagnosis of their first venous thromboembolism [69]. Although individuals with this particular illness have a higher likelihood of experiencing a first episode compared with the general population, their likelihood of experiencing a subsequent episode is equivalent to that of those with unprovoked deep vein thrombosis [70]. The elevated prevalence of spontaneous instances might perhaps be attributed to undetected thrombophilias [43].

The prevalence of venous thromboembolism is significantly elevated in persons of African American descent compared with those of White ancestry, with rates ranging from 30% to 100% higher [71]. The occurrence of deep vein thrombosis does not exhibit a gender preponderance. However, it has been shown that males have a higher likelihood of experiencing recurring thrombosis. The incidence of deep-vein thrombosis rises with progressive age, partly attributable to the higher frequency of medical ailments and other predisposing factors for thrombosis among the senior demographic [47]. Both smoking and obesity have been shown to be correlated with an increased susceptibility to deep-vein thrombosis [72].

## 5. Limitations

The research findings have significant implications and certain limitations that should be acknowledged. The observation that a significant proportion of individuals diagnosed with COVID-19 develop arterial thrombosis raises concerns regarding the efficacy of using anticoagulants as the sole preventive measure against thrombosis-related complications. The precise mechanism underlying thrombosis in COVID-19 has yet to be elucidated. 

The presence of hypoalbuminemia could potentially contribute to the development of thrombosis. However, in order to establish a definitive cause–effect relationship, further investigation is required through an interventional trial involving albumin supplementation. Moreover, there is a lack of information pertaining to the pre-existing nutritional status, which could potentially serve as a significant factor contributing to decreased levels of albumin. 

Ultimately, it is important to note that venous and arterial thrombosis may not possess identical pathogenetic mechanisms. Instead, they may manifest as distinct conditions that develop throughout the course of hospitalisation and carry varying risks for unfavourable outcomes. For instance, there may be a potential association between acute limb or cerebral ischaemic events and the presence of new-onset atrial fibrillation, which is frequently observed in cases of pneumonia. Additional research will be required in order to comprehensively comprehend the potential diverse ramifications associated with various types of vascular complications, as well as their distinct temporal occurrences throughout the progression of the disease.

## 6. Conclusions

In summary, this study contributes additional knowledge regarding the clinical manifestation of COVID-19, revealing the comparable occurrences of venous and arterial thrombosis. Individuals diagnosed with thrombosis experienced a more pronounced manifestation of the disease, accompanied by laboratory indicators of systemic inflammation and hypercoagulability. Episodes of ischaemia, as observed in this context, serve as a foreboding indication of diminished chances of survival. Consequently, the timely detection and implementation of suitable therapeutic measures for individuals at risk of COVID-19 could effectively address the thrombotic risk and mitigate mortality rates.

## Figures and Tables

**Figure 1 biomedicines-12-00774-f001:**
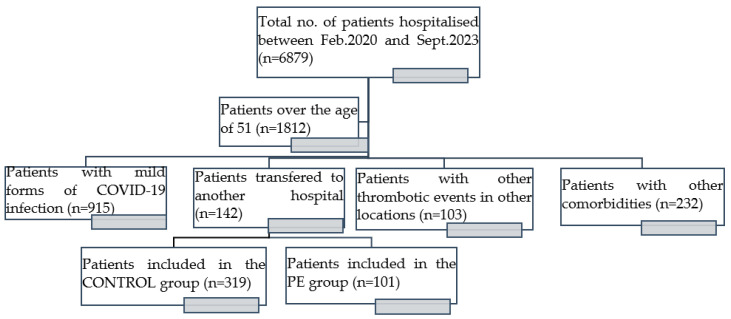
The steps in the recruitment process.

**Figure 2 biomedicines-12-00774-f002:**
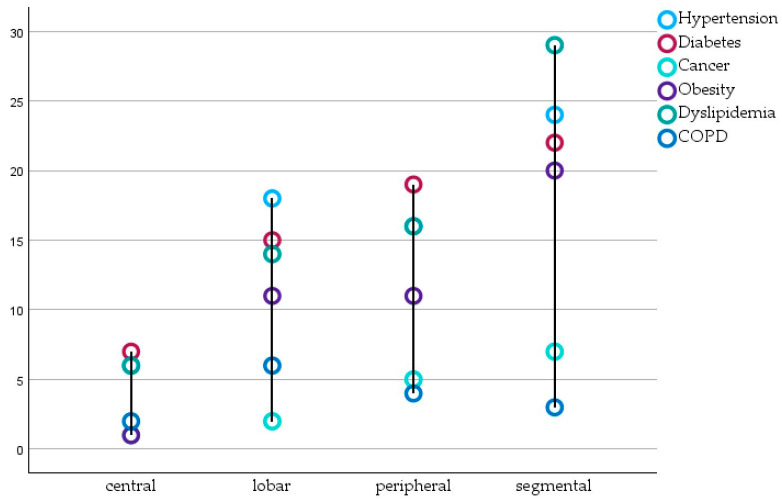
Patients with hypertension, COPD, cancer, obesity, diabetes and dyslipidaemia according to thrombus location.

**Figure 3 biomedicines-12-00774-f003:**
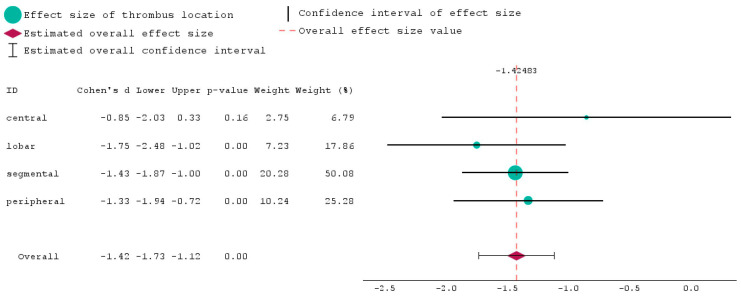
Forest plot of the thrombus locations.

**Figure 4 biomedicines-12-00774-f004:**
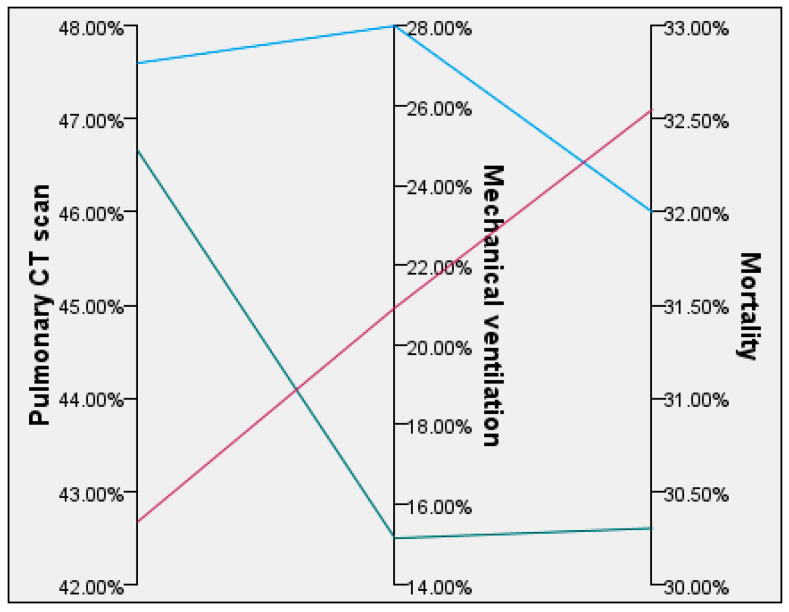
Relationship between pulmonary CT scan, mechanical ventilation, and mortality.

**Figure 5 biomedicines-12-00774-f005:**
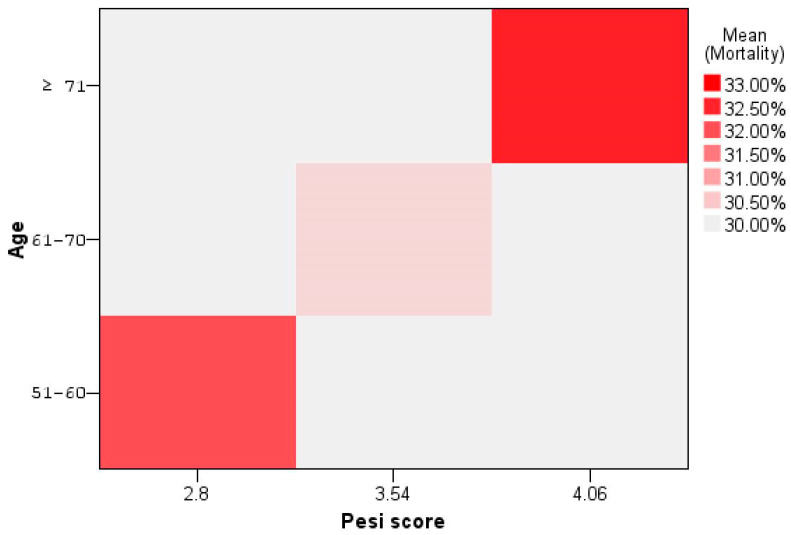
Relationship between age group, Pesi score, and mortality.

**Figure 6 biomedicines-12-00774-f006:**
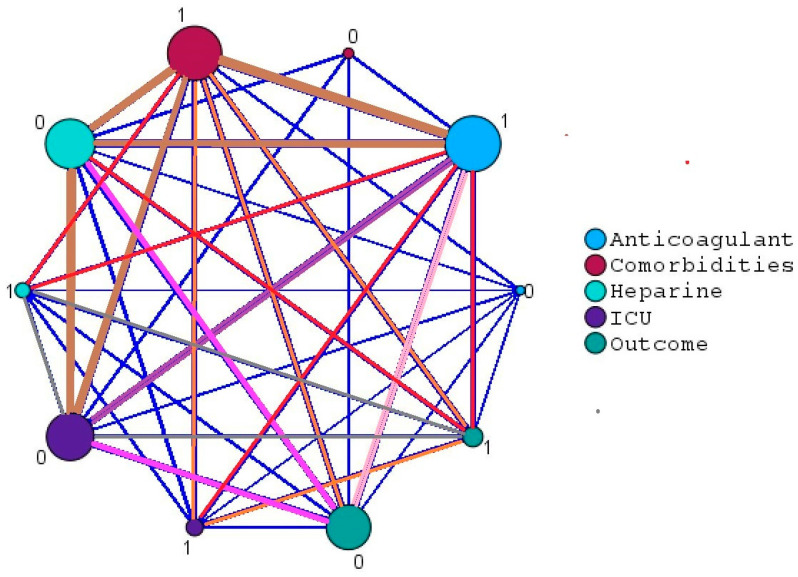
Relationship map between selected variables.

**Figure 7 biomedicines-12-00774-f007:**
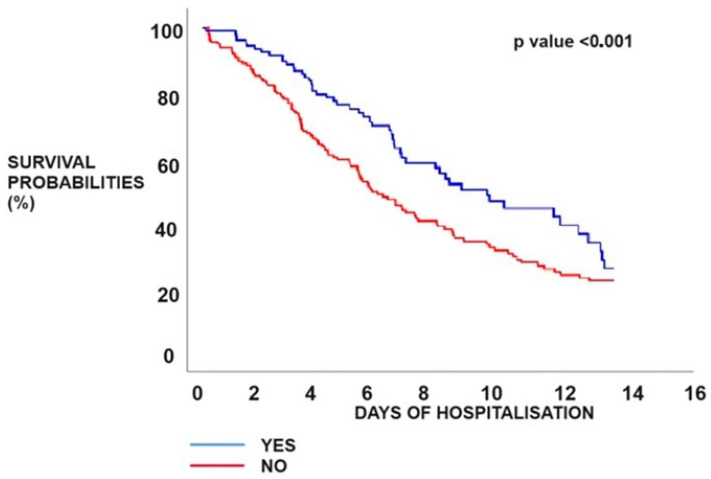
Kaplan–Meier analysis of mortality time in patients with or without thrombotic events during the in-hospital stay.

**Figure 8 biomedicines-12-00774-f008:**
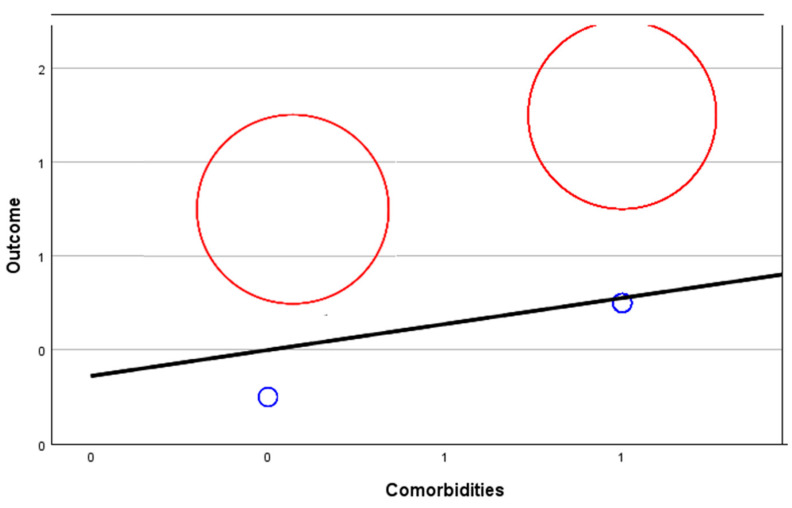
Meta-regression analysis of comorbidities in relation to the outcome of patients from the PE group (positive relationship).

**Figure 9 biomedicines-12-00774-f009:**
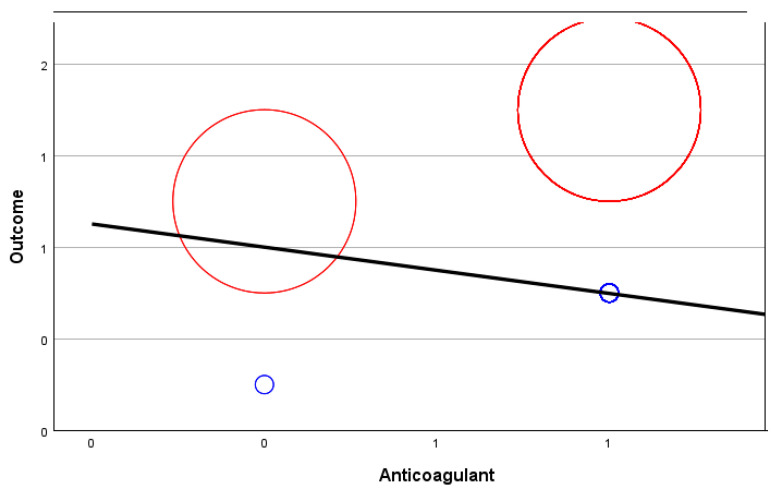
Meta-regression analysis of anticoagulant therapy in relation to the outcome of patients from the PE group (negative relationship).

**Figure 10 biomedicines-12-00774-f010:**
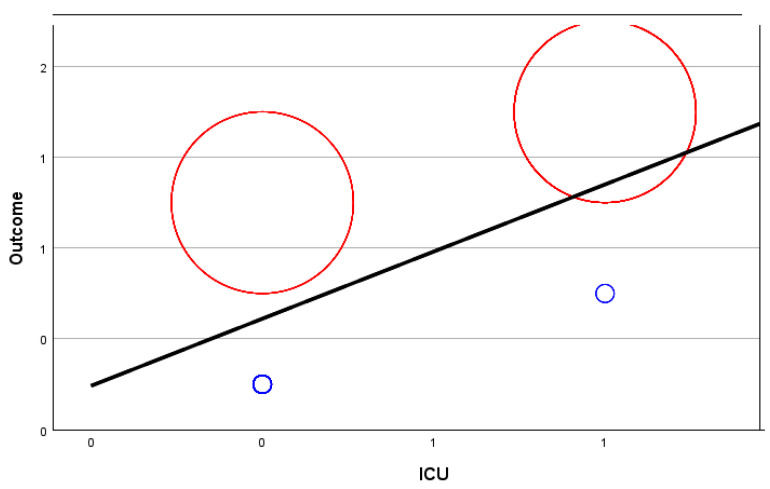
Meta-regression analysis of ICU admission in relation to the outcome of patients from the PE group (positive relationship).

**Figure 11 biomedicines-12-00774-f011:**
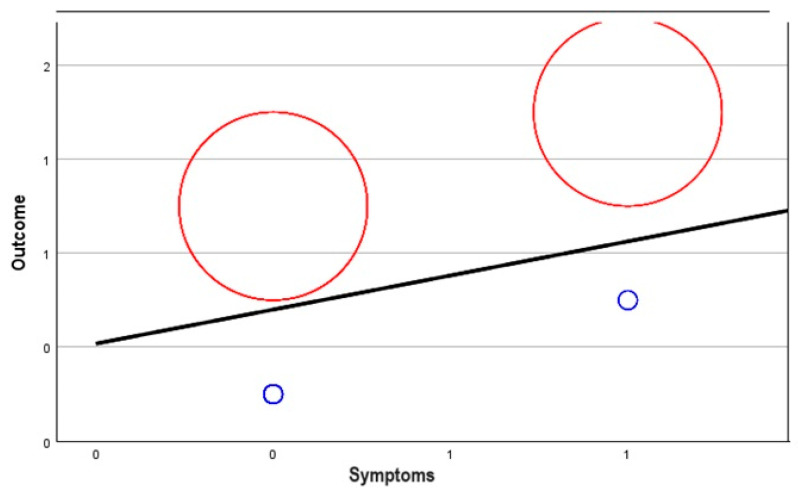
Meta-regression analysis of symptoms in relation to the outcome of patients from the PE group (positive relationship).

**Figure 12 biomedicines-12-00774-f012:**
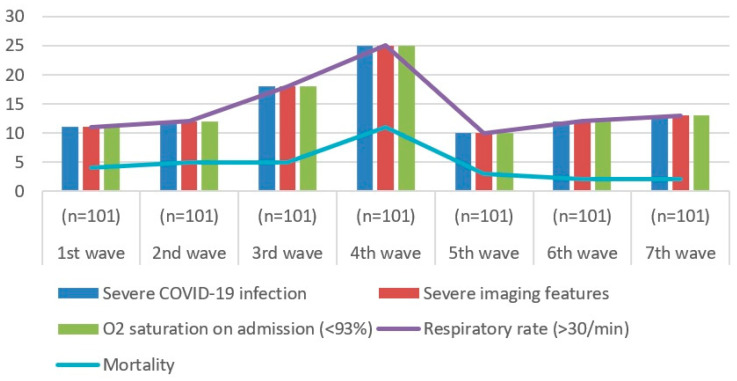
Comparison of pandemic waves for the PE group.

**Table 1 biomedicines-12-00774-t001:** Clinical and laboratory characteristics of study patients.

	All Patients	Control Group	PE Group	*p*-Value
Number of participants	420	319	101	
Age				
Mean ± SD	69.34 ± 16.09	67.47 ± 15.52	71.86 ± 13.27	0.329
Age group (%)				
51–60	124 (29.52%)	99 (31.03%)	25 (24.75%)	<0.001
61–70	138 (32.85%)	105 (32.91%)	33 (32.67%)	<0.001
≥70	158 (37.61%)	115 (36.05%)	43 (42.57%)	<0.001
Male sex	223 (53%)	168 (52.66%)	55 (55.45%)	<0.001
Geographical distribution				
Urban	271 (64.52%)	210 (65.83%)	61 (60.39%)	0.002
Rural	149 (35.47%)	109 (34.17%)	40 (39.61%)	0.026
COVID-19 vaccination status				
YES	194 (46.1%)	172 (53.9%)	22 (21.7%)	<0.001
NO	226 (53.8%)	147 (46.1%)	79 (78.2%)	<0.001
COVID-19 vaccine type				
Pfizer BioNTech	120 (61,86%)	106 (61.7%)	14 (63.6%)	0.007
Moderna	51 (26.29%)	46 (26.7%)	5 (22.7%)	0.006
Janssen	23 (11.85%)	20 (11.6%)	3 (13.7%)	0.002
ICU admission	45 (10.71%)	25 (7.8%)	20 (19.8%)	0.006
Comorbidities				
Hypertension	258 (61.42%)	194 (60.81%)	64 (63.36%)	0.032
Diabetes	169 (40.23%)	106 (33.22%)	63 (62.37%)	0.173
Smoker	215 (51.19%)	162 (50.78%)	53 (52.47%)	0.055
COPD	54 (12.85%)	39 (12.22%)	15 (14.85%)	0.021
CAD	70 (16.66%)	51 (15.98%)	19 (18.81%)	0.144
Heart failure	81 (19.28%)	57 (17.86%)	24 (23.76%)	0.071
Atrial fibrillation	53 (12.61%)	36 (11.28%)	17 (16.83%)	0.005
Dyslipidaemia	148 (35.23%)	105 (32.91%)	43 (42.57%)	0.012
BMI ≥ 30	205 (49.04%)	137 (42.94%)	69 (68.31%)	0.007
Cancer history	48 (11.42%)	35 (10.97%)	13 (12.87%)	0.014
Medication				
ACE-inhibitors	76 (18.09%)	51 (15.98%)	25 (24.7%)	0.031
ARBs	59 (14.04%)	48 (13.75%)	11 (10.89%)	0.053
Aspirin	53 (12.61%)	46 (14.42%)	7 (6.9%)	0.165
Statins	53 (12.61%)	31 (12.85%)	22 (21.78%)	0.002
Anticoagulant	62 (14.76%)	58 (18.18%)	4 (3.9%)	0.024
Antiplatelet agents	66 (15.71%)	64 (20.06%)	2 (1.9%)	0.081
Antihypertensives	204 (48.57%)	153 (47.96%)	51 (50.49%)	0.003
Diuretics	196 (46.66%)	148 (46.39%)	48 (47.5%)	0.006
Calcium channel blockers	215 (51.1%)	162 (50.78%)	53 (52.47%)	0.024
Antiarrhythmics	45 (10.71%)	31 (9.7%)	14 (13.86%)	0.161
Antidiabetics	158 (37.61%)	101 (31.66%)	57 (56.43%)	0.059
Remdesivir	338 (80.47%)	257 (80.56%)	81 (80.19%)	0.084
Favipiravir	66 (15.71%)	46 (13.47%)	20 (19.80%)	0.003
Tocilizumab	80 (19.04%)	67 (21%)	13 (12.87%)	0.005
Anakira	137 (32.61%)	102 (31.97%)	35 (34.65%)	0.071
Dexamethasone	276 (65.71%)	223 (69.9%)	53 (52.47%)	0.612
Actilyse	6 (1.4%)	0	6 (5.94%)	0.001
Nadroparin	203 (48.33%)	149 (46.7%)	54 (53.46%)	0.412
Heparins	60 (14.28%)	55 (17.24%)	5 (4.9%)	0.004
Enoxaparina	152 (36.19%)	129	23	0.053
PaO_2_/FiO_2_ [median]	313 [155–379]	328 [284–379]	254 [155–309]	<0.001
Laboratory findings (mean ± SD; [median])				
WBC (×1000/mm^3^)	8.9 [1–32]	8.5 [1.6–32]	9.6 [1–28]	0.344
PLT (×1000/mm^3^)	196 [11–682]	165 [11–440]	228 [30–682]	0.007
Limfocytes	974 [150–3462]	1100 [614–3462]	780 [150–2230]	0.249
Neutrofils	5967 [1087–27,330]	4422 [1087–11,452]	7745 [1700–27,330]	0.172
Creatinine (mg/dL)	1.71 ± 8.85	2.26 ± 11.77	1.01 ± 0.49	0.483
Total proteins	6 [3.3–7.3]	5.8 [5.1–7.3]	6.3 [3.3–7.1]	0.264
Na (mmol/L)	135 [0.78–154]	134 [0.78–154]	135.5 [125–149]	0.497
K (mmol/L)	4.18 [2.7–6.36]	4.21 [2.98–6.36]	4.1 [2.7–5.7]	0.410
Glycemia (mg/dL)	144 [57–501]	152 [57–501]	137 [68–423]	0.452
AST (U/L)	34 [8.8–321]	34 [8.8–321]	34 [11–213]	0.009
ALT (U/L)	38 [5–347]	37 [5–347]	44.5 [6–294]	0.007
GGT	65 [24–372]	68 [26–372]	66 [24–364]	0.055
Procalcitonin	0.4 [0.07–15]	0.58 [0.09–15]	0.3 [0.07–10]	0.013
CRP (mg/L)	90.5 [0.03–538]	70.5 [0.03–538]	114 [24–433]	0.029
ESR (mm/h)	76 [10–205]	78 [11–205]	75 [10–200]	0.162
D-dimer (ng/mL)	2765 [211–19,000]	2431 [211–5880]	3100 [790–19,000]	0.175
Ferritine (ng/mL)	884 [111–5600]	780 [111–4828]	890 [120–5600]	0.036
Fibrinogen	6.0 [4.79–8.35]	5.94 [4.79–6.91]	6.7 [4.8–8.35]	0.080
Albumin (g/dL)	3.51 [2.01–5.9]	3.3 [2.01–5.4]	3.7 [2.43–5.9]	0.009
IL6	77 [6–5600]	78 [6.6–5600]	76 [6–5500]	0.006
LDH units/L	348 [135–2634]	336 [135–2634]	391 [156–2405]	0.173
LDL	205 [78–376]	201 [78–301]	221 [91–376]	0.062
HDL	28 [22–50]	32 [24–50]	24 [22–49]	0.008
Triglycerides	224 [98–521]	213 [98–402]	265 [101–521]	0.005
Lactate	2.7 [0.4–4.7]	2.4 [0.4–4.1]	3.2 [0.6–4.7]	0.094
Cholinesterase	5224 [4288–9345]	5230 [4389–9345]	5211 [4288–9297]	0.274
Alkaline phosphatases	141 [50–162]	134 [50–151]	145 [75–162]	0.083
aPTT sec.	39.4 [26.5–60.2]	38.8 [26.5–51.1]	41.2 [27.3–60.2]	0.113
INR	1.11 [1.3–3.2]	1.9 [1.5–2.5]	2.4 [1.3–3.2]	0.244
Symptoms				
Digestive symptoms	45 (10.71%)	27 (8.46%)	18 (17.82%)	0.009
Anosmia	81 (19.28%)	65 (20.37%)	16 (15.84%)	0.005
Ageusia	102 (24.28%)	81 (25.39%)	21 (20.79%)	0.053
Fatigue	350 (83.33%)	274 (85.89%)	76 (75.24%)	0.061
Dyspnoea	234 (55.71%)	165 (51.72%)	69 (69.31%)	0.006
Confusion	26 (6.19%)	15 (4.7%)	11 (10.89%)	0.071
Headache	290 (70.47)	281 (88.08%)	15 (14.85%)	0.008
Fever	248 (59.04%)	179 (56.11%)	69 (68.3%)	0.031
Cough	266 (63.33%)	195 (61.12%)	71 (70.29%)	0.041
Clinical outcomes				
Moderate COVID-19	228 (54.29%)	198 (62.07%)	30 (29.70%)	<0.001
Severe COVID-19	192 (45.71%)	121 (37.93%)	71 (70.29%)	0.071
Severe imaging features (CT)	128 (30.4%)	88 (27.58%)	40 (39.6%)	0.003
Mean duration of hospital stay	13.4 ± 3.6	12.5 ± 3.1	14.1 ± 3.8	0.064
ICU admissions	46 (10.95%)	25 (7.8%)	21 (20.79%)	0.021
Mean duration from hospital admission to ICU admission	4.9 ± 2.1	5.1 ± 2.2	4.7 ± 1.3	0.025
Median duration of ICU stay	4.4 ± 1.9	4.1 ± 1.3	5.5 ± 2.1	0.031
Intubation	45 (10.71%)	25 (7.8%)	20 (19.8%)	0.006
Wells score	-	-	1.34 ± 1.68	<0.001
Pessi score	-	-	3.5 ± 1.03	<0.001
O_2_ supplementation	288 (68.57%)	204 (63.94%)	84 (83.16%)	0.322
Mortality	90 (21.42%)	58 (18.18%)	32 (31.68%)	<0.001

ICU: intensive care unit; WBC: white blood cells, PLT: platelets; CRP: C reactive protein; ACE: angiotensin-converting enzyme inhibitors; ARBs: angiotensin receptor blockers; CAD: coronary heart disease; COPD: chronic obstructive pulmonary disease; BMI: body mass index; ESR: erythrocyte sedimentation rate.

**Table 2 biomedicines-12-00774-t002:** Odds ratios of hypertension, COPD, cancer, obesity, diabetes and dyslipidaemia.

**Hypertension**		**Obesity**	
Odds ratio	1.1145	Odds ratio	2.8645
95% CI:	0.7015 to 1.7708	95% CI:	1.7828 to 4.6025
z statistic	0.459	z statistic	4.350
Significance level	*p* = 0.646	Significance level	*p* < 0.001
**Neoplasm**		**Diabetes**	
Odds ratio	1.1987	Odds ratio	3.3314
95% CI:	0.6073 to 2.3662	95% CI:	2.0923 to 5.3044
z statistic	0.522	z statistic	5.071
Significance level	*p* = 0.601	Significance level	*p* < 0.001
**COPD**		**Dyslipidemia**	
Odds ratio	1.2522	Odds ratio	3.6799
95% CI:	0.6585 to 2.3811	95% CI:	2.3013 to 5.8844
z statistic	0.686	z statistic	5.440
Significance level	*p* = 0.492	Significance level	*p* < 0.001

**Table 3 biomedicines-12-00774-t003:** Thrombus location.

	Thrombus Location				
	Central (*n* = 6)	Lobar (*n* = 20)	Segmental (*n* = 51)	Peripheral (*n* = 25)	*p*-Value
Age	66.5 ± 8.03	69.65 ± 9.24	70.44 ± 9.44	68.6 ± 10.06	<0.001
Male	50%	60%	51%	56%	-
Hypertension	9%	28.5%	37.5%	25%	-
COPD	13.3%	40%	20%	26.7%	-
Obesity	2.5	25.5%	46.5%	25.5%	-
Cancer	7%	13%	47%	33%	-
Diabetes	0	25.3%	49.4%	25.3%	-
Dyslipidaemia	9.3%	21.5%	44.6%	24.6%	-
PLT (×1000/mm^3^)	194 ± 111.6	237.7 ± 126.8	243.1 ± 122.4	238.2 ± 108.4	<0.001
D-dimers	4183.3 ± 2934.09	5888 ± 3411.92	4026.7 ± 3577.55	3126.9 ± 2317.5	0.033
Pesi	3.8 ± 1.32	3.75 ± 0.71	3.52 ± 1.14	3.52 ± 0.96	<0.001
Wells	1.6 ± 3.4	1.2 ± 1.15	1.35 ± 1.81	1.34 ± 1.24	0.160

**Table 4 biomedicines-12-00774-t004:** Mortality according to selected variables.

Age	Wells Score	Pulmonary CT Scan	Pesi Score	Mechanical Ventilation	Mortality
51–60 (*n* = 25)	1.1 ± 0.95	47.6%	2.8 ± 1.04	28%	32%
61–70 (*n* = 33)	0.8 ± 1.22	46.66%	3.54 ± 0.07	15.15%	30.3%
≥71 (*n* = 43)	1.89 ± 2.12	42.67%	4.06 ± 0.76	20.93%	32.55%
Overall (*n* = 101)	1.34 ± 1.68	45.19%	3.58 ± 1.03	20.79%	30.69%

**Table 5 biomedicines-12-00774-t005:** Correlations between different variables.

		Correlation	Count	Lower C.I.	Upper C.I.
Outcome	ICU	0.673	101	0.550	0.768
	Symptoms	0.563	101	0.114	0.470
	Comorbidities	0.161	101	−0.036	0.345
	Anticoagulant	−0.113	101	−0.301	0.085

**Table 6 biomedicines-12-00774-t006:** Matrix between selected variables.

Control Variables		Comorbidities	Anticoagulant	ICU	Symptoms
Comorbidities	Correlation	1.000	−0.038	0.038	0.743
	Significance *p*-value	-	0.707	0.711	0.463
	Degree of freedom (df)	0	98	98	98
Anticoagulant	Correlation	−0.038	1.000	0.067	0.165
	Significance *p*-value	0.707	-	0.506	0.102
	df	98	0	98	98
ICU	Correlation	0.038	0.067	1.000	0.814
	Significance *p*-value	0.711	0.506	-	0.814
	df	98	98	0	98
Symptoms	Correlation	0.743	0.165	0.814	1.000
	Significance *p*-value	0.463	0.102	0.425	-
	df	98	98	98	0

**Table 7 biomedicines-12-00774-t007:** Comparison of COVID-19 waves of the pandemic for the PE group.

	1st Wave (*n* = 101)	2nd Wave (*n* = 101)	3rd Wave (*n* = 101)	4th Wave (*n* = 101)	5th Wave (*n* = 101)	6th Wave (*n* = 101)	7th Wave (*n* = 101)
Severe COVID-19 infection	11	12	18	25	10	12	13
Severe imaging features	11	12	18	25	10	12	13
O_2_ saturation on admission (<94%)	11	12	18	25	10	12	13
Respiratory rate (>30/min)	11	12	18	25	10	12	13
Mortality	4	5	5	11	3	2	2

## Data Availability

The data presented in this study are available upon request from the corresponding author.

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
