# Peer review of "Risk Factors for Pulmonary Embolism in Individuals Infected with SARS-CoV2—A Single-Centre Retrospective Study"

_biomedicines, 2024, doi:10.3390/biomedicines12040774_

Round 1

Reviewer 1 Report

Comments and Suggestions for Authors

The proposed manuscript is devoted to the results of a research providing a comprehensive description of the characteristics and outcomes of 420 COVID-19 hospitalized patients in Romania, and their risk of developing pulmonary embolism.

Preliminaries to the research area are provided. Information related to the high risk of occurrence of thrombotic events in patients diagnosed with SARS-CoV-2 infection and possible cutaneous thrombosis, pulmonary embolism, stroke, or coronary thrombosis. Evidence suggesting that thromboembolism, hypercoagulability, and excessive production of proinflammatory cytokines, play a significant role in the development of multiorgan failure are reviewed.

The methodology of the proposed study is described in detail, in particular the study design, the fulfillment of ethical requirements, the inclusion and exclusion criteria for the patients, as well as the variables that were included for analysis. The results of the statistical analysis and their characteristics are presented. A detailed discussion of the results is given and corresponding conclusions are formulated.

The presentation of the main results is clear and comprehensive. From a formal point of view, all the contents seems to be correct. The results are valuable and worthy of being published taking into account their possible applications in clinical practice and health care.

Minor revisions are suggested to improve the quality of the exposition:

p. 5, line 218 and p. 14, line 351: The dashes at the beginning of the titles of Table 1 and Figure 7 should be removed.

p. 10, line 288: Here it should be the title of Figure 2 instead of the description of the contents of Figure 3.

p. 11, line 324. It should be “Overall (n=101)” instead of “Overall (n-101)”.

p. 13, line 343: I suggest to write “The thicker lines” instead of “The thincker lines”.

p. 23, line 695: The formatting of some tables, figures and the bibliography should be checked and improved, e.g. the number of the Ref. by Macanu et al. is incorrect.

Author Response

Dear Reviewer,

We would like to thank you very much for your valuable comments on our work and we really appreciate the time and effort you put into reading our research. We do hope that this updated version of the manuscript will add value to the existing knowledge about the risk factors for pulmonary embolism in individuals infected with SARS-CoV2, and that you will be satisfied with it. Please find below our comments point by point:

  1. p. 5, line 218 and p. 14, line 351: The dashes at the beginning of the titles of Table 1 and Figure 7 should be removed..

Thank you for your observation. We made the relevant changes.

  1. p. 10, line 288: Here it should be the title of Figure 2 instead of the description of the contents of Figure 3.

Thank you for your feedback and for pointing this out. We added the title of Figure 2, please see lines 320-321.

  1. p. 11, line 324. It should be “Overall (n=101)” instead of “Overall (n-101)”.

Thank you. We changed it.

  1. p. 13, line 343: I suggest to write “The thicker lines” instead of “The thincker lines”.

Thank you. We changed it.

  1. p. 23, line 695: The formatting of some tables, figures and the bibliography should be checked and improved, e.g. the number of the Ref. by Macanu et al. is incorrect.

Thank you for pointing this out. We checked again the bibliography and made the neccessary changes.

Also, regarding the English language, we have had this paper checked by our colleague, Dr. Andreea Nelson Twakor, who is also an official English translator, and we hope this is ok now. Please see below the translation authorisation.

Again, we would like to thank you for the feedback and, after taking your valuable recommendations into consideration, we made the necessary changes, and we hope you will find them appropriate and suitable for the journal. We also wish that this latest version will be a pleasant article to read.

Thank you again for your review and we look forward to hearing from you.

Dr. Alexandra Herlo

Reviewer 2 Report

Comments and Suggestions for Authors

The clinical study by Herlo et al performed some analyses of the risk factors for pulmonary embolism in SARS-CoV2-infected patients. Although the study is important and the results could have great impacts when the similar cases are found in some individuals, the overall quality (sample size, data, analytical methods, and interpretations) seems not qualified at the moment to be considered in this journal. This reviewer has some concerns for authors to be clarified in a possible revision.

1) The introduction is not good, especially for the readers to understand the background and the significance of the current study. The motivation and logic of this study are also not very clear. Moreover, there has been already several reports reviewed/studied the similar fields (Risk factor, health condition, primary illnesses). The authors should have done a better introduction. 

2) Sample size is one of the major concerns. How did the authors select those 420 individual cases? 

3) Gender factor was not included in the analysis? Why?

4) The control group used in this study is bit confusing. How and why those cased could be treated as CONTROL group?

5) Hypertension, Obesity, Neoplasm, Diabetes, COPD, Dyslipidemia are some major illnesses found high risk in this study. How about the "Stages" in those cases, which must have more information to describe the situation. A case-by-case study, considering ages and other co-factors might be also needed.

6) The interpretation for all figures are insufficient. It is hard to follow the article.

7) The overall organization and structure of the manuscript must be adjusted with only major findings presented in the main text.

Comments on the Quality of English Language

Needs extensive editing.

Author Response

Dear Reviewer,

Thank you very much for the opportunity to reconsider the manuscript and to undertake major revisions, marked with the colour red into the document, by addressing the observations received. We took into consideration your valuable comments on our work, and we really appreciate the time and effort you put into reading our research. We do hope that this updated version of the manuscript will add value to the existing knowledge about the risk factors for pulmonary embolism in individuals infected with SARS-CoV2, and that you will be satisfied with it. Please find below our comments point by point:

  • The introduction is not good, especially for the readers to understand the background and the significance of the current study. The motivation and logic of this study are also not very clear. Moreover, there has been already several reports reviewed/studied the similar fields (Risk factor, health condition, primary illnesses). The authors should have done a better introduction. 

We appreciate this comment and thank you so much for pointing this out. We do acknowledge that there is a bit of confusion with the Introduction and we changed/added some information to make it more understandable for the reader. We strongly agree with your observation, and we do appologise for this. We hope this is a better version of it and that you will agree to this.

  • Sample size is one of the major concerns. How did the authors select those 420 individual cases?

This is a very good point and we hope to be able to address this concern. In the study, we described the inclusion and exclusion criteria, as well as a flowchart describing the recruitment process (fig. 1). We selected the patients after careful consideration of the inclusion factors in order to ensure the relevance, feasibility, and ethical conduct of the research (Age 51 or older / Positive diagnosis of moderate or severe form of COVID-19, confirmed through a RT-PCR test / Patients were hospitalized due to the acute infection with SARS-CoV2, with at least one of the following comorbidities: hypertension, diabetes, COPD, CAD, heart failure, atrial fibrillation, dyslipidemia, BMI ≥ 30, cancer history / Patients developed pulmonary embolism / Patients were wither vaccinated or unvaccinated (disregarding what type was vaccine was performed).

We determined the required sample size based on statistical considerations, including the anticipated effect size, level of significance, power of the study, and expected dropout rates. We used various journal articles that describe how to select and what constitutes a good sample size:

  • We used Criterion sampling as described by Moser et al. - selection of participants who meet pre-determined criteria of importance (after extablishing the inclusion criteria we had 420 patients to build our data base with) - Moser A, Korstjens I. Series: Practical guidance to qualitative research. Part 3: Sampling, data collection and analysis. Eur J Gen Pract. 2018 Dec;24(1):9-18. doi: 10.1080/13814788.2017.1375091.
  • In our decision about sampling we used Phenomenology, in which (similar to the concept above), our participants had to meet a predefined criteria. The determining factor is the participant's criteria with the phenomena being investigated. We wanted our study patients to have collectively had an event (infection with SARS-CoV2), but different in terms of their features and individual experiences. For instance, our phenomenological investigation included individuals that tested positive for COVID-19 but with diverse characteristics such as age, comorbidities, existance of pulmonary thombembolism, development of a severe form of the infection - Owusu Sarfo, & Pritchard Debrah. (2021, December 5). Qualitative Research Designs, Sample Size and Saturation: Is Enough Always Enough? Journal of Advocacy, Research and Education, 8(3). https://doi.org/10.13187/jare.2021.3.60
  • From the final 420 patients included in the study, 101 had developed PE, and 319 had not. We based our statistical approach on this information and we analysed the clinical presentations and various factors that led to the development of PE

We really appreciate that you picked up on this and we hope that we answered your concern. To make it more understandable for our readers, we included suplementary information in the article, and we hope now that this explains better the recruitment process.  Should you have more concerns on this issue, please do let us know, so we can make the necessary adjustments.

  • Gender factor was not included in the analysis? Why?

Thank you for this question. We did not include gender in our initial recuirment process as we found strong evidence in the literature that pulmonary embolism tends to occurs equally in men and women. This is somehow proved by our remaining participants in the study, as we have 55.45% male that developed PE, as opposed to 44.55% females with the same condition. This 0.8% difference between the two genders is not enough (in our opinion) to support the idea that one gender is more prone to develop PE than the other one.  We hope this addressed your querry and that you agree with us on all the above. We included supporting information in the main text.

  • The control group used in this study is bit confusing. How and why those cased could be treated as CONTROL group?

Thank you for this. From the final 420 patients left (after applying the inclusion and exclusion criteria), only 101 developed PE. The remaining 319 were patients with severe forms of COVID-19. Since they did not had PE, we included them in the CONTROL group and we analysed the differenced between them and the PE group, trying to reach some conclusions as why they did not progress to PE. We hope this clarifies the question. We understand why it was somehow confusing, so we included in the main text more information on this.

  • Hypertension, Obesity, Neoplasm, Diabetes, COPD, Dyslipidemia are some major illnesses found high risk in this study. How about the "Stages" in those cases, which must have more information to describe the situation. A case-by-case study, considering ages and other co-factors might be also needed.

Thank you very much for this. We did not include the stages of these comorbidities because we intended to keep the article to a length that would be manageable for any reader. We do intend to write another article more focused on stages of these comorbidities in relation to thrombosis. But for this study, we prefered to keep it like this.

  • The interpretation for all figures are insufficient. It is hard to follow the article.

We understand and appreciate this. Thus, we added more information and changed the structure of the content to make it more understandable.

  • The overall organization and structure of the manuscript must be adjusted with only major findings presented in the main text.

Thank you very much for these observations. We have reconsidered the sections covering all these issues above and explained in more details. We have reconfigured the Discussion section, we moved part of the text to results and added new paragraphs and more studies on this topical subject in the scientific literature, by keeping sight also on the references.

Regarding the quality of the English language, we have had this paper checked by our colleague, Dr. Andreea Nelson Twakor, who is also an official English translator, and we hope this is ok now. Please see below the translation authorisation.

Again, we would like to thank you for the feedback and, after taking your valuable recommendations into consideration, we made the necessary changes, and we hope you will find them appropriate and suitable for the journal. We also wish that this latest version will be a pleasant article to read.

Thank you again for your review and we look forward to hearing from you.

Thank you again

Dr. Alexandra Herlo

Reviewer 3 Report

Comments and Suggestions for Authors

I checked only the statistical part. I left the medical part to the specialists. In my opinion, the statistical study was basically done correctly. I have only a few comments regarding the tables and the bibliography provided.

(1) The description of Figure 3 next to the drawing is virtually illegible.It needs to be fixed.

(2) Figure 6. I suggest that "Relationship count" be described with lines of different colors.The thickness of the lines in the main drawing is difficult to distinguish.

(3) Figure 7. It is better to put "survival probabilities" on the vertical axis. Then the picture will be more understandable.

(4) The reference list should be organized according to the MDPI style.

Author Response

Dear Reviewer,

We would like to thank you very much for your valuable comments on our work and we really appreciate the time and effort you put into reading our research. We do hope that this updated version of the manuscript will add value to the existing knowledge about the risk factors for pulmonary embolism in individuals infected with SARS-CoV2s, and that you will be satisfied with it. We are very pleased that you appreciated the Statistics section. Please find below the answers to your comments:

  1. The description of Figure 3 next to the drawing is virtually illegible.It needs to be fixed.

Thank you so much for pointing this out. We fixed it.

  1. Figure 6. I suggest that "Relationship count" be described with lines of different colors.The thickness of the lines in the main drawing is difficult to distinguish.

Thank you for your suggestion. We changed the colours and it looks better indeed.

(3) Figure 7. It is better to put "survival probabilities" on the vertical axis. Then the picture will be more understandable.

We agree with this recommendation. Thank you so much, we changed it to “survival probabilities”

(4) The reference list should be organized according to the MDPI style.

We revised the reference list accordingly.

Regarding the English language, we have had this paper checked by our colleague, Dr. Andreea Nelson Twakor, who is also an official English translator, and we hope this is ok now. Please see below the translation authorisation.

Again, we would like to thank you for the feedback and, after taking your valuable recommendations into consideration, we made the necessary changes, and we hope you will find them appropriate and suitable for the journal. We also wish that this latest version will be a pleasant article to read. Thank you again for your review and we look forward to hearing from you.

Dr. Alexandra Herlo

Round 2

Reviewer 2 Report

Comments and Suggestions for Authors

This reviewer has no further comments on this revised manuscript.

Comments on the Quality of English Language

Fine